# Photodynamic Therapy against Colorectal Cancer Using Porphin-Loaded Arene Ruthenium Cages

**DOI:** 10.3390/ijms251910847

**Published:** 2024-10-09

**Authors:** Suzan Ghaddar, Aline Pinon, Manuel Gallardo-Villagran, Jacquie Massoud, Catherine Ouk, Claire Carrion, Mona Diab-Assaf, Bruno Therrien, Bertrand Liagre

**Affiliations:** 1Faculté de Pharmacie, Univ. Limoges, LABCiS, UR 22722, F-87000 Limoges, France; suzan.ghaddar@etu.unilim.fr (S.G.); aline.pinon@unilim.fr (A.P.); villagran@outlook.com (M.G.-V.); jacquie.massoud@hotmail.com (J.M.); 2Doctoral School of Sciences and Technology, Lebanese University, Beirut 21219, Lebanon; mdiabassaf@ul.edu.lb; 3Institut de Chimie, Université de Neuchâtel, Avenue de Bellevaux 51, CH-2000 Neuchâtel, Switzerland; 4Univ. Limoges, CNRS, Inserm, CHU Limoges, BISCEm, UAR 2015, US 42, F-87000 Limoges, France; catherine.ouk@unilim.fr (C.O.); claire.carrion@unilim.fr (C.C.)

**Keywords:** colorectal cancer, photodynamic therapy, photosensitizers, arene-ruthenium assemblies, apoptosis

## Abstract

Colorectal cancer (CRC) is the third most common cancer in the world, with an ongoing rising incidence. Despite secure advancements in CRC treatments, challenges such as side effects and therapy resistance remain to be addressed. Photodynamic therapy (PDT) emerges as a promising modality, clinically used in treating different diseases, including cancer. Among the main challenges with current photosensitizers (PS), hydrophobicity and low selective uptake by the tumor remain prominent. Thus, developing an optimal design for PS to improve their solubility and enhance their selective accumulation in cancer cells is crucial for enhancing the efficacy of PDT. Targeted photoactivation triggers the production of reactive oxygen species (ROS), which promote oxidative stress within cancer cells and ultimately lead to their death. Ruthenium (Ru)-based compounds, known for their selective toxicity towards cancer cells, hold potential as anticancer agents. In this study, we investigated the effect of two distinct arene-Ru assemblies, which lodge porphin PS in their inner cavity, and tested them as PDT agents on the HCT116 and HT-29 human CRC cell lines. The cellular internalization of the porphin-loaded assemblies was confirmed by fluorescence microscopy. Additionally, significant photocytotoxicity was observed in both cell lines after photoactivation of the porphin in the cage systems, inducing apoptosis through caspase activation and cell cycle progression disruptions. These findings suggest that arene-Ru assemblies lodging porphin PS are potent candidates for PDT of CRC.

## 1. Introduction

Colorectal cancer (CRC) is the third most common cancer and the second leading cause of cancer-related deaths worldwide. Annually, it accounts for about 1.85 million diagnoses and 850,000 deaths [1]. The primary treatments include standard systemic chemotherapy, immunotherapy, and alternative therapies that target the mechanisms responsible for promoting cancer spread and progression [2]. In recent decades, substantial advancements in treating CRC have been made through the introduction of innovative medications and treatment strategies. However, an augmentation in the resistance of tumor cells combatting these therapies is a widespread phenomenon, and most of these therapies do not show selective targeting of cancer cells, which harms healthy tissues and generates a variety of adverse effects [3,4,5,6]. Therefore, the development of innovative drugs with high efficiency and selectivity has become a must.

Photodynamic therapy (PDT) has undergone significant historical development, beginning with observing photodynamic effects in the early 20th century. The pivotal discovery of hematoporphyrin derivatives in the 1960s laid the groundwork for PDT’s therapeutic applications. Subsequent clinical trials culminated in the approval of PDT by the U.S. Food and Drug Administration (FDA) in 1993 for treating cancers, making it a viable cancer treatment modality [7,8,9]. The mechanism of PDT revolves around the interaction of photosensitizers (PS), light, and molecular oxygen to induce cytotoxic effects in target tissues. Upon light absorption at specific wavelengths [10], PS undergoes excitation and transfers energy to molecular oxygen, generating reactive oxygen species (ROS). These ROS, particularly singlet oxygen, initiate a cascade of biochemical reactions leading to cellular damage and eventually cell death. The key principle of PDT lies in its ability to selectively target tumor cells while sparing adjacent healthy tissues [11].

CRC, as a prevalent malignancy, presents an area of interest for investigating the efficacy of PDT through the evaluation of novel PS on human CRC cell lines [12]. In this context, the PDT action mechanism should be well understood. This process starts when the PS is activated through the absorption of photons from light allowing its transformation from a singlet ground state to a short-lived electronically excited singlet state. At this stage, the PS is very unstable and can decay back to its ground state by losing excess energy through light emission and/or heat production. On the other hand, this excited singlet state can also undergo intersystem crossing to develop into a more stable long-lived electronically excited triplet state, which can directly transfer energy into molecular oxygen (O_2_) through type I and/or type II photochemical reactions. In the type I process, the excited PS reacts with biological substrates such as cell membranes to give free radicals and radical ions through hydrogen atom abstraction or electron transfer. These species can undergo a reaction with O_2_ and stimulate the production of superoxide anion, hydroxyl radical, or hydrogen peroxide. The kinetics of both reactions are highly dependent on the availability of oxygen, the nature of the PS, and the substrate concentration, as well as occurring simultaneously, leading to the reversion of the excited state into its ground state. Indeed, ROS are highly cytotoxic and can oxidize diverse biomolecules through the induction of redox reactions, causing oxidative stress and subsequent cell death [13,14,15].

Among PS used in PDT, tetrapyrrole derivatives including porphyrins, chlorins, bacteriochlorins, and phthalocyanines are known to have low solubility in water and biological media, forcing their aggregation and generating improper photophysical properties, thus limiting their selectivity, and overall, their attractiveness in healthcare modalities [16]. Therefore, to enhance the PS absorption rate and bioavailability, drug delivery systems are often used [17]. This includes vectorization through nanoparticles that can selectively transport the PS into the tumor sites by passive diffusion due to the enhanced permeability and retention (EPR) effect. The EPR effect is an outcome of increased vessel permeability, poor lymphatic drainage, and leaky vasculature. Vectorization can also augment the hydrophilicity of PS, allowing their proper characterization in water and increasing their availability in blood [18,19,20]. Lodging of PS within carriers was elaborated through the development of the third-generation PS to overcome the limitations associated with the first- and second-generation PS [21]. Such a strategy has been previously used by us, showing that arene-Ru metalla-assemblies can act as drug delivery vectors [22,23,24].

In parallel, metal-based chemotherapeutic agents including cisplatin, carboplatin, and oxaliplatin are frequently used to treat cancer. In this context, Ru-based complexes are also used and showed better overall efficiency as compared to the other metal-based agents, because they can facilitate more selective targeting to tumor sites. Essentially, Ru ions can bind to transferrin proteins, which accordingly bind to transferrin receptors that are highly expressed on cancer cells [25,26,27]. This highlights the advantage of Ru derivatives in chemotherapy since the ruthenium toxicity, antiangiogenic, antimetastatic, and overall effects on cellular processes are well documented [28,29]. Therefore, using coordination-driven self-assembled arene-Ru cages to carry the PS to cancer cells is quite efficient, as demonstrated in previous studies [22,23,24].

Herein, the biological effect of two prismatic arene-Ru water-soluble organometallic cages (**M1** and **M2**) encapsulating 21*H*, 23*H*-porphin (**PS**) within their cavity was evaluated (Figure 1). This encapsulation yielded **PS⸦M1** and **PS⸦M2** host–guest systems (Figure 2), which can be internalized into cancer cells after the Ru-transferrin-mediated binding to transferrin receptors. Once internalized, the PS gets excited at specific wavelengths of light illumination to induce cellular oxidative stress, cycle arrest, and apoptosis.

The lodging of porphin inside their cavities was designed to enhance the solubility of porphin in biological media, with porphin being the simplest porphyrin-based PS. However, porphin alone is not soluble in water, like most porphyrinic compounds [30], and therefore, the use of organometallic cages was essential for this study. The organometallic cages, [(ր^6^-*p*-cymene)_6_Ru_6_(donq)_3_(tpt)_2_][CF_3_SO_3_]^6+^ (**M1**) and [{(ր^6^-*p*-cymene)Ru}_6_(dotq)_3_(tpt)_2_][CF_3_SO_3_]^6+^ (**M2**), were built from three main components, including the arene-Ru vertices, the panels (tpt = 2,4,6-tris(pyridin-4-yl)-1,3,5-triazine), and the spacers (donq = 5,8-dioxido-1,4-naphthoquinonato; dotq = 6,11-dioxido-5,12-naphthacenedionato) that act as edges. The anti-inflammatory activity of these assemblies after PDT was previously evaluated for the first time in vitro on a primary, non-cancerous rheumatoid arthritis fibroblast-like synoviocytes (RA-FLS) cell line, which mainly focused on the influence of PDT on key mediators of the inflammatory process, without delving into the PDT-induced cell death mechanism [31]. In contrast, our study shifts the focus entirely to the anticancer potential of these assemblies by evaluating their ability to induce photocytotoxicity and cell death, specifically in a CRC model, which represents a distinctly different disease type. To establish their therapeutic efficacy, we assessed the effect of these assemblies on two human CRC cell lines of different stages, HCT116 and HT-29, and thoroughly investigated their ability to generate ROS, induce apoptotic cell death, and alter the cell cycle progression through various biological assays. HCT116 cells are known to be less resistant than HT-29 cells due to their less differentiated state, which allows this study to encompass the investigations on more than one stage of CRC development [32].

## 2. Results

### 2.1. Cell Viability and Phototoxic Effect Evaluation

To investigate the effects of our arene-Ru assemblies in vitro, we conducted a series of experiments on HCT116 and HT-29 human CRC cell lines. The cell lines were treated with the compounds **PS⸦M1** and **PS⸦M2**, as well as the empty assemblies **M1** and **M2**, which lack the **PS** component in their inner cavity. Following treatment, the cells were either exposed to red light illumination (630 nm and 75 J/cm^2^) or kept in the dark. The impact on cell viability was assessed at 12 h, 24 h, and 48 h post-illumination using the MTT assay, a standard method for measuring cellular metabolic activity as an indicator of cell viability and proliferation. The results of the MTT assay revealed that in the absence of light, there was not any significant cytotoxic effect exhibited on the HCT116 and HT-29 cell lines after treatment with **PS⸦M1**, **PS⸦M2, M1**, or **M2** (> 1000 nM). This indicates that these compounds are non-toxic to cells under dark conditions, ensuring that any observed cytotoxicity can be attributed to photoactivation rather than intrinsic toxicity. However, upon exposure to red light, a marked decrease in cell viability was observed for both **PS⸦M1 and PS⸦M2 systems** (Figure 3). This phototoxic effect was both dose- and time-dependent, with significant reductions in cell viability noted at 12 h, 24 h, and 48 h post-illumination for both HCT116 and HT-29 cell lines (Figure 3A,B). Specifically, the IC_50_ values, representing the concentration at which 50% of the cells are inhibited, were found to be in the nanomolar range for both the **PS⸦M1** and **PS⸦M2** host–guest systems, indicating the high potency of these compounds under light activation. For the HCT116 cell line, the IC_50_ determined for **PS⸦M1** at 12 h, 24 h, and 48 h post-illumination were 666 nM, 602 nM, and 588 nM, respectively. Moreover, the IC_50_ values determined for **PS⸦M2** at 12, 24, and 48 h post-illumination were 647 nM, 541 nM, and 518 nM, respectively. For the HT-29 cell line, the IC_50_ values of **PS⸦M1** were 892 nM, 672 nM, and 536 nM, and for **PS⸦M2**, were 884 nM, 593 nM, and 539 nM at 12 h, 24 h, and 48 h post-illumination, respectively (Figure 3C). We notice that the determined IC_50_ values of **PS⸦M1** at each time post-illumination were slightly greater than those of **PS⸦M2** in both the HCT116 and HT-29 cell lines. In illuminated conditions, the phototoxicity was significantly high, whereas in the dark, the IC_50_ values could not be determined, underscoring the lack of cytotoxicity without light activation. Interestingly, the individual components **M1** and **M2**, when not carrying porphin, did not exhibit significant changes in cell viability under illuminated or non-illuminated conditions (Figure 3A,B). This indicates that the full assemblies **PS⸦M1** and **PS⸦M2** are crucial for the observed photodynamic activity facilitating the targeted delivery and effective photoactivation of the PS. In summary, these results demonstrate that the **PS⸦M1** and **PS⸦M2** assemblies possess significant phototoxic activity, leading to substantial reductions in cell viability upon light exposure. For subsequent experiments, the IC_50_ values determined after illumination are used to further explore the therapeutic potential and mechanistic aspects of the compounds.

### 2.2. ROS Production

The efficacy of PDT is directly linked to the production of ROS, a crucial aspect. To assess ROS levels, the cells were stained with the dichlorodihydrofluorescein (DCFDA) substrate and analyzed using flow cytometry immediately after illumination, with hydrogen peroxide (H_2_O_2_) being used as a positive control. The results revealed a substantial increase in ROS production following illumination in the treated cells compared to the treated non-illuminated and control cells. Specifically, in the HCT116 cell line, treatment with the **PS⸦M1** assembly under light exposure resulted in a remarkably high level of ROS production of 88.46%, significantly exceeding the 12.79% observed in the absence of light (Figure 4A). Similarly, the **PS⸦M2** assembly induced a notable ROS level of 89.20% under light exposure, compared to 16.79% in the dark. Parallel findings were observed in the HT-29 cell lines, with the ROS levels reaching 83.02% and 87.68% for **PS⸦M1** and **PS⸦M2**, respectively, under light exposure. These values were significantly higher than the 6.03% and 7.40% observed in the absence of light for **PS⸦M1** and **PS⸦M2**, respectively (Figure 4B). These results underscore the photoactivation of the **PS⸦M1** and **PS⸦M2** assemblies through ROS production, affirming their role in exerting the PDT effect. The substantial increase in ROS levels following illumination provides compelling evidence of the effectiveness of these assemblies in generating cytotoxic ROS species, validating their potential as promising candidates for PDT applications.

### 2.3. Cellular Internalization

The molecular uptake and internalization of the **PS⸦M** assemblies within cells were determined by confocal microscopy. After the internalization of the metallacages into the cells by endocytosis, the guest molecule was released from the host, allowing it to be excited by light and to undergo photoactivation [33]. Porphin, which is inherently auto-fluorescent in the red or infrared regions, can be efficiently tracked after its internalization into the cells [34]. After treatment with **PS⸦M1** and **PS⸦M2** assemblies, the fluorescence was predominantly observed in the cytoplasm of the HCT116 and HT-29 cell lines. This cytoplasmic accumulation confirms the cellular uptake of these assemblies, which potentially permits their interaction with cytosolic targets (Figure 5A,B). Indeed, this cytoplasmic localization is crucial for understanding the intracellular distribution and potential mechanisms of action of these assemblies.

### 2.4. Cell Cycle Analysis

As cancer cells possess a rapid proliferation rate, it is important to investigate the effects of the **PS⸦M** assemblies on the cell cycle progression. The number of cells present at each phase of the cell cycle was monitored through DNA labeling with propidium iodide (PI). The effect of the **PS⸦M1** and **PS⸦M2** assemblies on cell cycle progression was examined in the HCT116 and HT-29 cell lines. The distribution of cells across different phases of the cell cycle was analyzed by flow cytometry (Figure 6A,B). In the HCT116 cell line, the sub-G1 phase, indicative of apoptotic cells, was 8.43% after 12 h of illumination with **PS⸦M1** and 8.23% with **PS⸦M2**, which were higher than that of the control cells (1.48%). At 24 h post-illumination, there was a significant increase in the sub-G1 phase for the treated and illuminated cells, rising to 21.42% for **PS⸦M1** and 34.34% for **PS⸦M2** as compared to the control cells (0.60%). Similar trends were observed at 48 h post-illumination, with sub-G1 phase cells reaching 24.14% for **PS⸦M1** and 36.67% for **PS⸦M2** as compared to the control cells (3.66%) (Figure 6C). The data indicate that the sub-G1 phase culminates at 24 h post-illumination, with no significant change observed at 48 h compared to 24 h. Furthermore, there was a noticeable accumulation of cells in the G1 phase with a decrease in the number of cells passing into the S-phase in the illuminated cells compared to the control cells, which was more prominent at 12 h and 24 h post-illumination. These findings suggest that the maximum phototoxic effect of the **PS⸦M1** and **PS⸦M2** assemblies in the HCT116 cell line is reached at 24 h post-illumination, with a pronounced increase in apoptosis and G1 phase arrest contributing to this effect. In the HT-29 cell line, at 12 h post-illumination, the sub-G1 phase was not observed; nevertheless, at 24 h post-illumination, the emergence of the sub-G1 phase was evident, with 3.96% of cells observed with **PS⸦M1** and 2.49% with **PS⸦M2** as compared to 0.03% in the control cells. The proportion of cells in the sub-G1 phase increased significantly at 48 h post-illumination, reaching 11.02% for **PS⸦M1** and 12.28% for **PS⸦M2** (Figure 6D). This delayed onset of the sub-G1 phase highlights a time-dependent apoptotic response to the treatment contributing to a progressive accumulation of apoptotic cells over time. Additionally, a persistent accumulation of cells in the G2 phase was observed at 12 h, 24 h, and 48 h post-illumination accompanied by a significant decrease in the number of cells entering the G1 phase. This indicates that these assemblies induce a cell cycle arrest in the G2 phase of the HT-29 cell line, blocking the ability of the cells to proceed to mitosis. In both cell lines, the assemblies do not influence the overall cell cycle progression in the dark. However, after illumination, we can conclude that the difference indicates that HT-29 cells have a slower apoptotic response compared to HCT116 cells, potentially due to variations in their cellular machinery or sensitivity to treatments.

### 2.5. Annexin V-PI

To investigate the apoptotic mechanisms as suggested by the sub-G1 phase of the cell cycle, we conducted an Annexin V-FITC/PI dual staining assay, analyzed via flow cytometry. This assay distinguishes early apoptotic cells, marked by Annexin V after membrane flip flop and phosphatidylserine exposure, as well as late apoptotic and necrotic cells, identified by PI uptake due to membrane integrity loss. In the HCT116 and HT-29 cell lines, the number of cells that were positively stained with Annexin V and PI increased significantly following the treatment with **PS⸦M1** and **PS⸦M2** assemblies after illumination, as compared to the control (Figure 7A,B). In the HCT116 cell line, at 12 h post-illumination, the percentage of apoptotic cells (early + late apoptosis) was 17.64% for **PS⸦M1** and 19.81% for **PS⸦M2**, indicating an early onset of apoptosis. This increased dramatically at 24 h post-illumination to 34.38% for **PS⸦M1** and 49.05% for **PS⸦M2**, nearly tripling the apoptotic response observed in non-illuminated treated cells (11.96% and 14.50%, respectively). By 48 h, apoptosis rates were 23.90% for **PS⸦M1** and 36.37% for **PS⸦M2**, still elevated compared to the non-illuminated cells of 9.07% and 10.98%, respectively, but lower than the 24 h peak, indicating a maximal apoptotic activity around 24 h post-illumination (Figure 7C). In the HT-29 cell line, the apoptotic response was also evident but exhibited a distinct pattern. At 12 h post-illumination, the percentage of apoptotic cells was 19.04% for **PS⸦M1** and 21.59% for **PS⸦M2**. At 24 h post-illumination, apoptosis increased to 31.68% for **PS⸦M1** and 34.13% for **PS⸦M2** while remaining relatively constant in the non-illuminated cells, 7.81% and 11.86%, respectively. At 48 h post-illumination, the percentage of apoptotic cells further increased to 40.12% for **PS⸦M1** and 41.56% for **PS⸦M2**, whereas the non-illuminated groups showed minimal apoptosis, 6.11% and 6.95%, respectively (Figure 7D). These results demonstrate that these PS assemblies induce significant apoptotic cell death in the HCT116 and HT-29 cell lines following illumination. The apoptotic response in the HCT116 cells peaks at 24 h post-illumination, while the HT-29 cells exhibit a more gradual increase in apoptosis, reaching a significant level at 48 h. The observed apoptotic responses confirm the photocytotoxic effects of these porphyrin-based PS assemblies, supporting their potential use in PDT.

### 2.6. Western Blot

To evaluate the onset of apoptosis, the expression of different proteins was analyzed by Western blot. During apoptosis, a complex cascade of events is initiated within cells, culminating in the activation of various apoptotic proteins, including caspase 3. Caspase 3 is initially synthesized in its inactive form, pro-caspase 3, which can become activated after a proteolytic cleavage. Once activated, caspase 3 will target and cleave poly (ADP-ribose) polymerase-1 (PARP-1). The cleavage of PARP-1 by active caspase 3 impairs the enzyme’s ability to repair DNA, effectively signaling the cell to proceed with apoptosis. Hence, the level of expression of caspase 3 and PARP-1 in the HCT116 and HT-29 cell lines was evaluated after photoactivation or not of the **PS⸦M** assemblies (Figure 8A,B). Our analysis revealed that in both cell lines, the activation of caspase 3, indicated by the presence of its cleaved form, was detected exclusively in cells that were subjected to illumination after treatment with **PS⸦M1** and **PS⸦M2**. In contrast, the cells that were kept in the dark and the control cells exhibited no sign of caspase 3 cleavage. This activation pattern was consistent at both 24 h and 48 h post-illumination, designating the induction of apoptosis in response to our treatment. Similarly, we observed that the cleavage of PARP-1, which serves as a hallmark of apoptosis, occurred only in the treated and illuminated cells. This cleavage was evident at both 24 h and 48 h time points, correlating with the observed activation of caspase 3. This highlights the potential of the assemblies as effective agents for inducing apoptosis through a targeted, light-dependent mechanism. The consistent findings in both the HCT116 and HT-29 cell lines further validate the general applicability of our approach.

### 2.7. DNA Fragmentation

To further elucidate the apoptotic mechanisms, we assessed nuclear changes through DNA fragmentation using ELISA assay in both cell lines. In the HCT116 cell line, there was a marked increase in DNA fragmentation 24 h post-illumination. The levels of DNA fragmentation increased by 4.7-fold for **PS⸦M1** and by 6.2-fold for **PS⸦M2** compared to the control cells. By 48 h post-illumination, the fold increase was 3.1 for **PS⸦M1** and 4.4 for **PS⸦M2**, suggesting a peak at 24 h, followed by a slight decline (Figure 9A). For the HT-29 cell line, a significant rise in DNA fragmentation was observed 24 h post-illumination. The levels increased by 2.5-fold for **PS⸦M1** and by 2.9-fold for **PS⸦M2** compared to the control cells. At 48 h post-illumination, DNA fragmentation further increased to 4.2-fold for **PS⸦M1** and 5.6-fold for **PS⸦M2**, indicating a sustained and progressively increasing apoptotic response (Figure 9B). For both cell lines, no significant increase in DNA fragmentation was observed at 12 h post-illumination after treatment with either **PS⸦M1** or **PS⸦M2**. Additionally, non-illuminated cells treated with **PS⸦M1** or **PS⸦M2** did not exhibit any significant changes in DNA fragmentation levels at 12 h, 24 h, and 48 h compared to the control cells. These findings underscore the role of **PS⸦M1** and **PS⸦M2** in inducing DNA fragmentation as part of the apoptotic process, with the most significant changes observed at 24 h post-illumination for HCT116 cells and a progressive increase up to 48 h for HT-29 cells. The lack of significant DNA fragmentation in non-illuminated cells and at earlier time points highlights the specificity and timing of the apoptotic response triggered by these PS assemblies upon illumination.

## 3. Discussion

As conventional anticancer therapies are often associated with numerous undesirable side effects, PDT came as a breakthrough in cancer therapy for its effectiveness and selectivity towards the tumor site. Only a few PS have been approved for medical applications, with porphyrin-based compounds being the most prominent [35]. However, this class of PS has a low solubility in biological media, which often constrains their use, applications, and implications in different studies. Fortunately, this limitation can be managed through the use of delivery vectors that encapsulate the PS so that a facilitated delivery and effective internalization by cells can be reached. To attain this outcome, the conjugation of porphyrin with metal-based drugs is considered a strategic approach for the treatment of several cancer types including CRC [36]. Ru complexes exhibit distinctive photochemical and photophysical properties conferring their uniqueness among metal-based therapies such as their ability to interact with cellular biomolecules and selectively accumulate in cancer cells [34,37]. As reported in previous studies, Ru has grabbed the attention of many researchers as a modality of treatment of various cancer types including prostate, colorectal, bladder, and hepatic cancer due to its ability to act as a chemotherapeutic cytotoxic agent with selectivity to the cancer cells [22,23,24,38,39].

In this research, we examined the anticancer efficacy of arene-Ru assemblies lodging porphin PS within their cavity on HCT116 and HT-29 CRC cell lines. Our findings demonstrated the potency of the Ru metallacages (**M1** and **M2**) to act as delivery systems of the porphin since they did not exhibit a cytotoxic effect under the studied concentration range either in the presence or in the absence of illumination. Hence, the IC_50_ of the **PS⸦M** systems determined with light illumination after different times, ranging between 500 and 900 nM, is mainly due to the porphin PS rather than the metallacage. In this study, the main difference between our **PS⸦M** systems was in the spacer acting as the edges of the cage and joining the Ru ions. In the study of Gallardo-Villagrán et al., a correlation between the spacer size and the release of porphin PS was proved through IC_50_ determination. In their study, the same metallacages with porphin PS inside the cavity were investigated on rheumatoid arthritis (RA) synoviocytes, and the IC_50_ of the **PS⸦M2** (named G1⸦M3 in their study) was around two-fold lower than that of **PS⸦M1** (named G1⸦M2 in their study) at 18 h post-illumination. This was linked to the fact that a bulkier spacer made up of four aromatic rings (**M2**) allows for the facilitated release of PS from the internal cavity of the cage as compared to that with two aromatic rings (**M1**). Additionally, the spacer that was made up of only one aromatic ring had a significantly higher IC_50_ value compared to both **PS⸦M1** and **PS⸦M2** [31]. However, in a previous study, the latter spacer was also included in the same cage with porphin PS inside and investigated on DU 145 and PC-3 prostate cancer cell lines, and the determined IC_50_ values were found to be in nanomolar ranges between 500 and 650 nM after 24 h of illumination [23]. These values were close to our findings with **PS⸦M1** and **PS⸦M2** treatments on HCT116 and HT-29 CRC cell lines. We also notice that in our study, **PS⸦M2** with the larger spacer has a slightly better overall effect than **PS⸦M1**, validating the study of Gallardo- Villagrán et al. [31], yet the difference is not significant. Here, we can demonstrate that the cell lines originating from different diseases will ultimately possess different sensitivities to the same treatment, also taking into consideration the impact of the difference in the drug–light time intervals. Even the different cell lines originating from the same disease will rationally not display similar outcomes. This was confirmed in our study when the HCT116 and HT-29 CRC cell lines showed a different overall outcome to the same treatment. As HCT116 is more sensitive than HT-29, the IC_50_ values obtained in our study with the same treatment at 12 h post-illumination were about two-fold lower than that of HT-29. Additionally, the observed apoptotic effect was reached faster in the HCT116 than in the HT-29 cell line where the maximal effect of the complexes was obtained 24 h post-illumination in the HCT116 cell line, whereas a relatively similar effect was obtained 48 h post-illumination on the HT-29 cell line. This was confirmed by the apoptotic phenomenon observed through Annexin V-PI staining and DNA fragmentation levels relative to time. DNA fragmentation can be enhanced when the PS is chemically modified or conjugated to facilitate its binding and interaction with DNA, as evidenced by the study of Zhao et al. [40]. Similarly, a study conducted by Rani-Beeram et al. demonstrated that fluorinated porphyrin-Ru showed a significant interaction with DNA, causing their cleavage in melanoma cells [41]. Additionally, the apoptotic induction of HepG2 liver cancer cells after photoactivation of the Ru porphyrin complex Rup-03 has been previously established [42]. Apoptotic cell death was further confirmed by Western blot through the cleavage of pro-caspase 3 and PARP-1 proteins. In the dark, there was a total absence of cytotoxicity in the treatment, which is also consistent with many previous studies [20,22,23,24,31]. Furthermore, we assessed the progression of the cell cycle and observed a peak for sub-G1, marking the cell death after photoactivation in both cell lines. Another study by Li et al. established a successful apoptotic effect on A549 lung carcinoma cells, distinguished by a sub-G1 peak after treatment with several Ru complexes [43]. Also, it was demonstrated that the cell cycle blockage was not the same on both cell lines; as mentioned earlier, the studied systems caused an arrest of the cell cycle at the G1 phase in the HCT116 cell line and the G2 phase in HT-29. Another study showed that Ru tetrapyridylporphin assemblies investigated on CRC caused a blockage in the S phase of the cell cycle [22]. As our systems are intrinsically auto-fluorescent, their intracellular accumulation was tracked by confocal microscopy, and image acquisitions showed that they are localized in the cytoplasm of the cells. Previous studies mentioned that the PS can be released from the cage either through a partial/ total rupture of the cage or through an aperture, which is the case in our study. It was also demonstrated that these cages enter the cell to deliver their contents through the assessment of the Ru ions inside the cells by inductively coupled plasma mass spectrometry (ICP-MS) [33,34]. After internalization into the cells, the released PS gets excited in the cytoplasm at specific wavelength (630 nm) and thereby contributes to detrimental effects on cells through the production of ROS. The cytoplasmic localization of Ru assemblies including PS, as well as ROS production triggering apoptosis, was also recognized in earlier investigations [22,23]. Overall, our results agree with the study of Lu et al., which demonstrated the anticancer potential of Ru complexes on hepatocellular carcinoma through growth inhibition of the HCC cells, induction of apoptosis marked by the sub-G1 peak, activation of caspases, and DNA fragmentation [39].

Further studies to elaborate on the involved mechanisms that these systems follow to release PS after internalization as well as the involved kinetics would be remarkable. Also, performing pre-clinical research on these systems through advanced culturing and in vivo models would be a principal objective for their further use as treatment options in clinics as they hold promising potential as candidates for PDT agents.

## 4. Materials and Methods

### 4.1. Materials

RPMI 1640 medium, RPMI 1640 red-phenol-free medium, fetal bovine serum (FBS), L-glutamine, and penicillin-streptomycin were purchased from Gibco BRL (Cergy-Pontoise, France). 3-(4,5-dimethylthiazol-2-yl)-2,5-diphenyltetrazolium bromide (MTT), cell death detection enzyme-linked immunosorbent assay (ELISA)^PLUS^, 2′,7″-dichlorofluoresceine diacetate (DCFDA), and human anti-β-actin antibody were obtained from Sigma-Aldrich (Saint-Quentin-Fallavier, France). Pro-caspase 3, cleaved caspase 3, poly-ADP-ribose polymerase-1 (PARP-1) antibodies, and goat anti-rabbit IgG secondary antibody conjugated to horseradish peroxidase (HRP) were acquired from Cell Signaling Technology-Ozyme (Saint-Quentin-en-Yvelines, France). Rabbit anti-mouse IgG-IgM H&L HRP secondary antibody, Annexin V-FITC, and PI were obtained from Invitrogen-Thermo Fisher Scientific (Villebon-Sur-Yvette, France). Immobilon Western Chemiluminescent HRP Substrate was acquired from Merck (Lyon, France).

### 4.2. Synthesis of the Arene-Ruthenium Assemblies Lodging a Photosensitizer

The assemblies were synthesized according to the literature [31]. The PS, *21*H, *23*H-porphin (**PS**), was synthesized as reported in the literature [44], and the corresponding lodging into arene Ru assemblies was attained by host–guest interactions. Stock solutions were prepared at 1 mM concentration in DMSO and stored at −20 °C.

### 4.3. Cell Culture and Treatment

The human CRC cell lines HCT116 and HT-29 were purchased from the American Type Culture Collection (ATCC-LGC Standards, Molsheim, France). The cells were grown in RPMI 1640 medium for culturing and were supplemented with 10% FBS, 1% L-glutamine, 100 U/mL penicillin, and 100 µg/mL streptomycin. The cultures were maintained in a humidified atmosphere containing 5% CO_2_ at 37 °C. For all experiments, the cells were seeded at 1.2 × 104 and 2.1 × 104 cells/cm^2^ for the HCT116 and HT-29 cells, respectively. The stock solution **PS⸦M1** and **PS⸦M2** assemblies were diluted in a culture medium to obtain the final concentration required, and the medium was replaced by a red phenol-free culture medium before PDT. The maximal concentration of DMSO in the culture medium was less than 0.1% in all cases, which is non-toxic.

### 4.4. Light Source

Illumination of the cells was achieved using a Lumidox^®^ II device (Analytical Sales and Services, Flanders, NJ, USA) under red visible light at 630 nm, at 75 J/cm^2^, and 70 mW/LED for 6 min. 

### 4.5. Cell Viability Assay

The cell viability was assessed using an MTT reagent (Sigma-Aldrich) as an indicator of the metabolic activity of the cells. The cells were seeded in 96-well culture plates and incubated for 24 h before treatment with the **PS⸦M1** and **PS⸦M2** assemblies, which were diluted in RPMI 1640 medium to obtain an appropriate range of concentration. After 24 h, the cells were illuminated or not using a Lumidox lamp 70 mW/LED. MTT assays were performed at 12 h, 24 h, and 48 h post-illumination, and cell viability was expressed as a percentage of cell viability for each treatment condition normalized to the untreated cells.

### 4.6. Intracellular ROS Generation

Cellular ROS generated after the illumination of the cells was quantified using a DCFDA substrate (Sigma-Aldrich). The cells were seeded in 75 cm^2^ culture flasks for 24 h; then, they were treated or not with **PS⸦M1** and **PS⸦M2** assemblies at their determined IC_50_ values. After 24 h, the cells with different treatment conditions were collected and incubated with DCFDA for 30 min before illumination. The quantification of cellular ROS production was determined directly after illumination through CytoFLEX LX (Beckman Coulter, Brea, CA, USA). H_2_O_2_ was used as a positive control at 800 μM concentration.

### 4.7. Cellular Internalization

The internalization of the compounds within the HCT116 and HT-29 cells was determined by confocal microscopy. The cells were seeded into incubation chambers (ibidi μ-Slide 8 wells; Clinisciences, Martinsreid, Germany) with a density of 2.5 × 10^4^ and 3.5 × 10^4^ cells/well for the HCT116 and HT-29 cell lines, respectively. The wells were coated with a gel prepared by mixing collagen type I (3 mg/mL) and acetic acid (20 mM) and incubated for 24 h before their treatment or not with **PS⸦M1** and **PS⸦M2** at their IC_50_ values. After treatment, image acquisitions were taken by a Carl Zeiss LSM 510 Meta―×1000 confocal laser microscope. The natural fluorescence of the assemblies permitted their observation within the cells with emission (excitation: 405/561 nm; emission: 650/673 nm).

### 4.8. Cell Cycle Analysis

The distinct phases of the cell cycle progression were assessed for both the HCT116 and HT-29 cell lines by flow cytometry using PI staining. The cells were seeded into 75 cm^2^ flasks and incubated for 24 h. The treatment or not with **PS⸦M1** and **PS⸦M2** assemblies at their determined IC_50_ values was performed after 24 h of incubation. Illumination of the cells was carried out after 24 h, and then, pellets of 1 × 10^6^ cells were collected at 12 h, 24 h, and 48 h post-illumination. The cells were washed with PBS and fixed by the addition of 1 mL of chilled 70% ethanol in PBS and stored at −20 °C. Before flow cytometry analysis, the cell pellets were washed with cold PBS and centrifuged, and then, they were resuspended with 500 µL of cold PBS along with 30 µL of RNase A (10 mg/mL) and incubated for 30 min at room temperature. After incubation, the cells were stained with 25 µL of PI (1 mg/mL), allowing the determination of the percentage of cells present in each phase of the cell cycle using the FACs system (BD Biosciences, San Jose, CA, USA).

### 4.9. Apoptotic Assays

#### 4.9.1. Annexin V-FITC/PI Dual Staining Assay

The cells were seeded into 75 cm^2^ flasks and were treated or not based on the determined IC_50_ values of the **PS⸦M1** and **PS⸦M2** assemblies after 24 h of incubation. After 24 h, the cells were illuminated or not and were harvested by trypsin at 12 h, 24 h, and 48 h post-illumination. Cell pellets of 2.5 × 10^5^ for each condition were prepared, washed in PBS, centrifuged, and resuspended in 300 µL of Annexin V lysis buffer (1X). The cells were labelled with 5 µL Annexin V-FITC and 1 µL of PI (0.1 mg/mL) and then were kept in the dark and at room temperature for 15 min. The percentage of cells undergoing apoptosis was then determined by CytoFLEX LX (Beckman Coulter).

#### 4.9.2. Protein Extraction and Western Blot Analysis

Cells of different conditions were gathered to extract and identify specific proteins of interest. After cell seeding for 24 h, the cells were treated or not with **PS⸦M1** and **PS⸦M2** assemblies at the determined IC_50_ values and collected 12 h, 24 h, and 48 h post-illumination. For total protein extraction, the cells were lysed with RIPA lysis buffer containing protease inhibitors according to the manufacturer’s instructions, as described [20]. The protein levels were quantified using the Bradford method; then, were separated on 12.5% SDS-PAGE gels; and then, transferred to PVDF membranes (Amersham Pharmacia Biotech, Saclay, France). Apoptosis-related human proteins were studied by membrane probing with primary anti-PARP-1 (1:1000), primary anti-cleaved caspase 3 (1:1000), and primary anti-pro-caspase 3 (1:1000). For the loading control, primary human anti-β-actin (1:5000) was used. After incubation with the primary antibodies overnight at 4 °C, appropriate secondary antibodies were used to incubate the membranes for at least 1 h. The blots were revealed using an Immobilon Western Chemiluminescent HRP substrate in a G: BOX system (Syngene, Cambridge, UK).

#### 4.9.3. DNA Fragmentation

A Cell Death ELISA^PLUS^ kit was used to assess the level of DNA fragmentation in the different cell conditions. After the cells were seeded in 75 cm^2^ flasks for 24 h, the cells were treated or not using the IC_50_ values of the **PS⸦M1** and **PS⸦M2** assemblies. After 24 h, the cells were illuminated or not and cell pellets of 2 × 10^5^ cells were collected for each condition. The DNA fragmentation levels were assessed according to the manufacturer’s protocol [22] and expressed as a fold change compared to the control cells.

### 4.10. Statistical Analysis

All data are expressed as the mean ± standard error of the mean (SEM) of three independent experiments. The statistical significance of the results was evaluated by a two-tailed unpaired Student’s *t*-test and expressed as * *p* < 0.05, ** *p* < 0.01, and *** *p* < 0.001.

## 5. Conclusions

In this in vitro study, we investigated the effects of arene-Ru assemblies with porphin sitting in their cavity (**PS⸦M**) on the HCT116 and HT-29 human CRC cell lines. The two host–guest systems showed relatively similar effects on both cell lines, with **PS⸦M2** exhibiting a slightly greater efficacy, possibly due to its bulkier structure facilitating the release of the PS. After being internalized into the cell cytoplasm, the assemblies demonstrated significant phototoxicity, primarily through the production of ROS, leading to oxidative stress, cell cycle arrest, and apoptosis. These findings suggest the potential use of these assemblies in anticancer PDT.

## Figures and Tables

**Figure 1 ijms-25-10847-f001:**
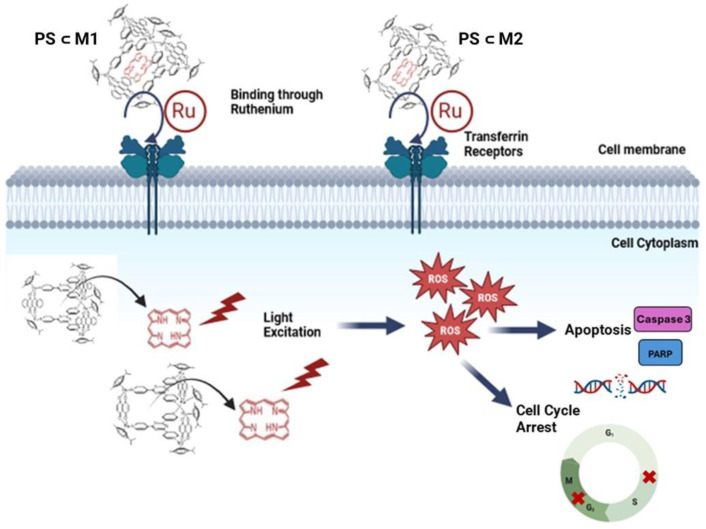
Schematic description of the cellular mechanisms induced by **PS⸦M1** and **PS⸦M2**, evaluated in this study. (Created with BioRender.com (accessed on 29 June 2024)).

**Figure 2 ijms-25-10847-f002:**
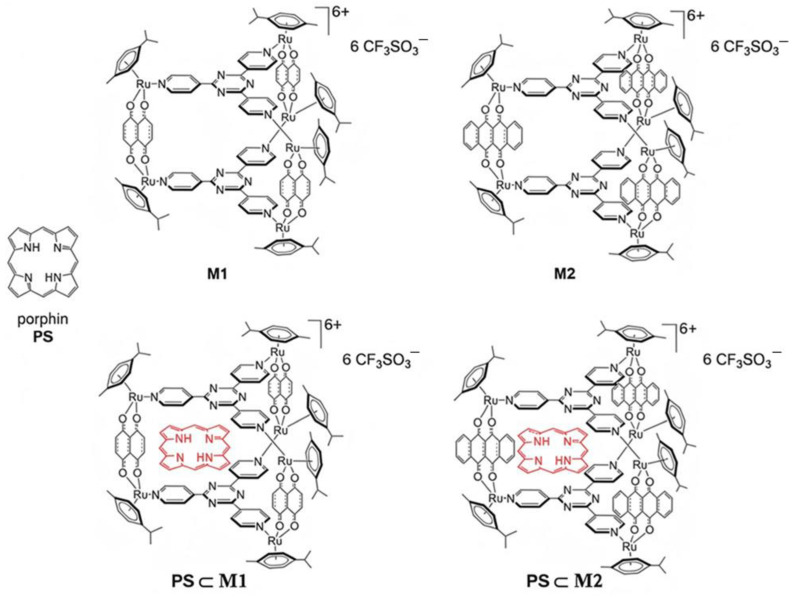
Chemical structure of porphin (**PS**) and the arene-ruthenium assemblies **M1** and **M2**, together with the host–guest systems **PS⸦M**.

**Figure 3 ijms-25-10847-f003:**
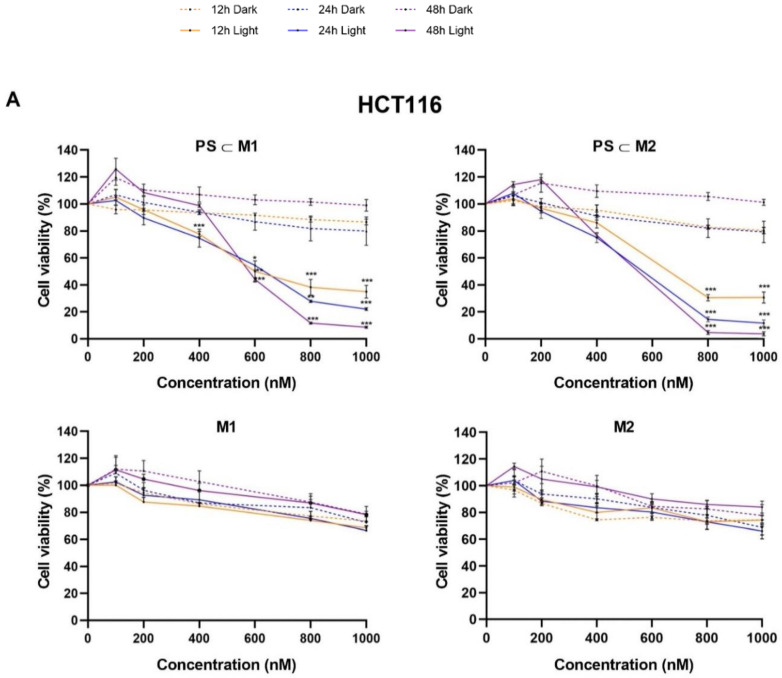
Phototoxicity of arene-Ru porphin **PS** assemblies on human CRC cell lines. (**A**) HCT116 and (**B**) HT-29 cell lines were cultured in RPMI medium for 24 h. After 24 h, cells were treated or not with **PS⸦M1**, **PS⸦M2**, **M1**, or **M2**. Illumination (630 nm, 75 J/cm^2^) of the cells occurred 24 h after treatment, and the cell viability for all the conditions was determined 12 h, 24 h, and 48 h post-illumination. Data are represented as a mean ± SEM of three independent experiments. * *p* < 0.05; ** *p* < 0.01; and *** *p* < 0.001. (**C**) Graphical representation of the IC_50_ values (nM) of **PS⸦M1** and **PS⸦M2** determined by MTT assay on HCT116 and HT-29 cell lines. Data are represented as a mean ± SEM of three independent experiments.

**Figure 4 ijms-25-10847-f004:**
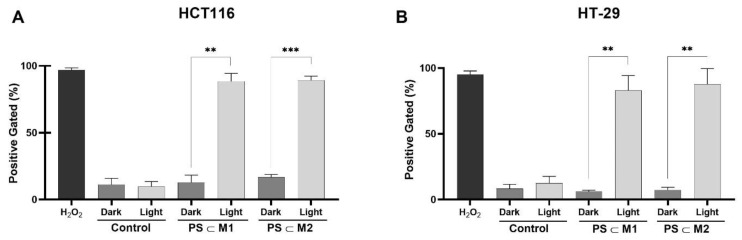
Intracellular ROS production was evaluated in (**A**) HCT116 and (**B**) HT-29 human CRC cell lines immediately after illumination (630 nm, 75 J/cm^2^) using DCFDA staining. Cells were analyzed using flow cytometry. Quantification of the intensity of fluorescence emitted due to DCF formation is correlated to the level of ROS generation. Data are represented as a mean ± SEM of three independent experiments. ** *p* < 0.01 and *** *p* < 0.001.

**Figure 5 ijms-25-10847-f005:**
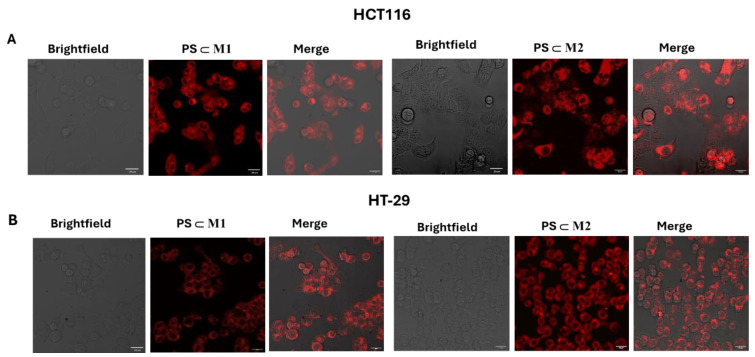
Detection of the cellular internalization of **PS⸦M1** and **PS⸦M2** in HCT116 (**A**) and HT-29 (**B**) cell lines by confocal microscopy. The cells were seeded into incubation chambers and cultured for 24 h. The cells were then treated with the compounds, and the fluorescence was measured by confocal microscopy (laser Zeiss LSM 510 Meta―×1000). The internalization was processed using the ImageJ image-processing software (version 1.54f). White scale bar = 20 μm.

**Figure 6 ijms-25-10847-f006:**
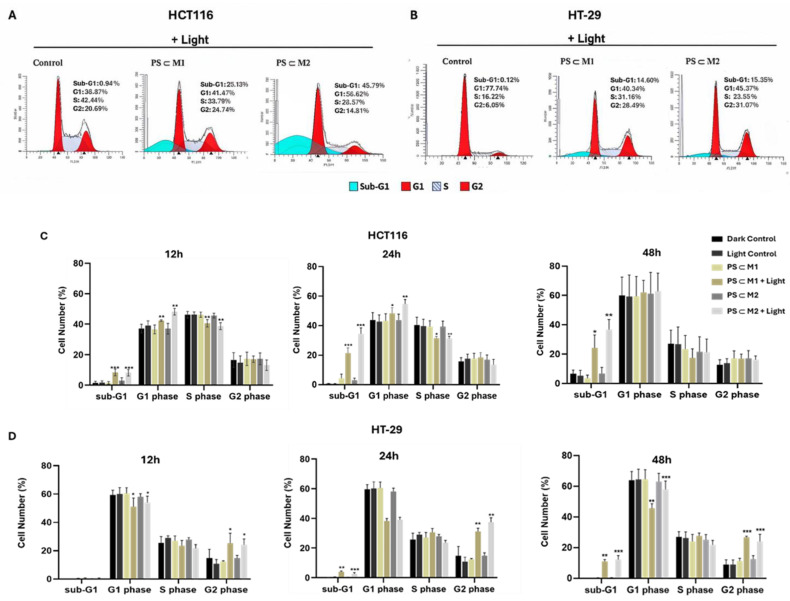
Cell cycle distribution analysis on HCT116 and HT-29 cell lines after the photoactivation of **PS⸦M1** and **PS⸦M2**. Cells were seeded for 24 h in a culture medium before the treatment or not with the assemblies at their determined IC_50_ concentrations. After 24 h of treatment, cells were illuminated or not with red light at 630 nm and 75 J/cm^2^. Cells were collected at 12 h, 24 h, and 48 h post-illumination for analysis by flow cytometry using PI staining. Images of the cell cycle distribution on (**A**) HCT116 cells at 24 h post-illumination and (**B**) HT-29 cell line at 48 h post-illumination are represented. Histograms representing the percentage cell numbers at each phase of the cell cycle on (**C**) HCT116 and (**D**) HT-29 cell lines are displayed as a mean ± SEM of three independent experiments. * *p* < 0.05; ** *p* < 0.01 and *** *p* < 0.001.

**Figure 7 ijms-25-10847-f007:**
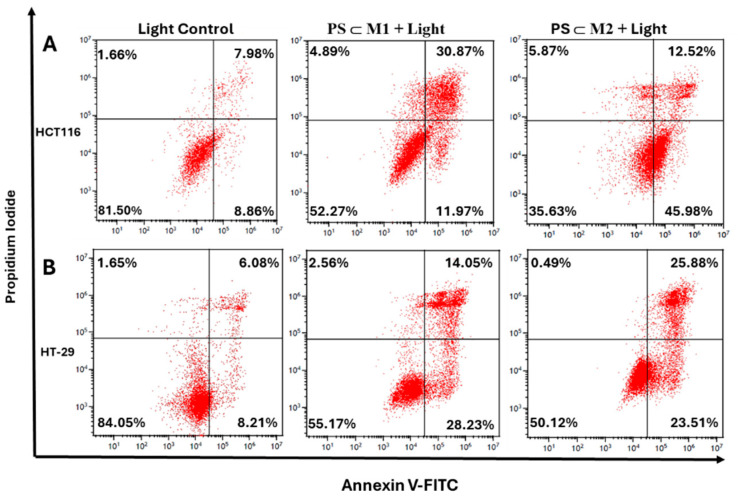
Apoptosis due to photoactivation was assessed on HCT116 and HT-29 cell lines. Cells were seeded and incubated for 24 h then treated or not with **PS⸦M1** and **PS⸦M2** assemblies at IC_50_ concentrations. Cells were either illuminated (630 nm, 75 J/cm^2^) or not after 24 h of treatment, and then, they were collected at 12 h, 24 h, and 48 h post-illumination. The collected cells were stained with Annexin V- FITC and PI, and their state was revealed by flow cytometry. Representative data from flow cytometry for the HCT116 cell line at 24 h post-illumination (**A**) and HT-29 at 48 h post-illumination (**B**) are displayed. Histograms represent the viable and apoptotic cell percentages of the treated HCT116 (**C**), and HT-29 (**D**) cells subjected to prior illumination. Data are represented as a mean ± SEM of three independent experiments. * *p* < 0.05; ** *p* < 0.01; and *** *p* < 0.001.

**Figure 8 ijms-25-10847-f008:**
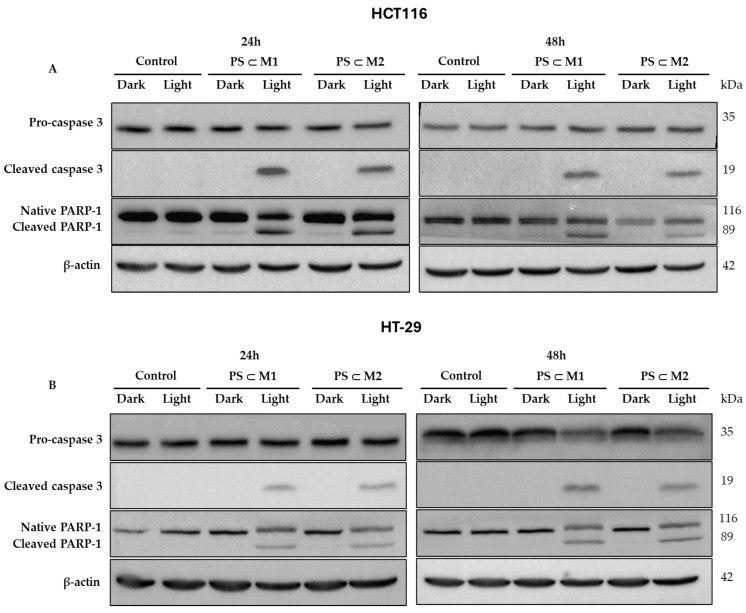
Protein expression was evaluated by Western blot. HCT116 (**A**) and HT-29 (**B**) cells were seeded and incubated for 24 h and then were treated or not at IC_50_ values of **PS⸦M1** or **PS⸦M2**. After treatment, cells were either illuminated (630 nm, 75 J/cm^2^) or not and collected 24 h and 48 h post-illumination. Proteins were extracted, and the level of protein expression of the different conditions were revealed. β-actin was used as a loading control. Representative images are shown.

**Figure 9 ijms-25-10847-f009:**
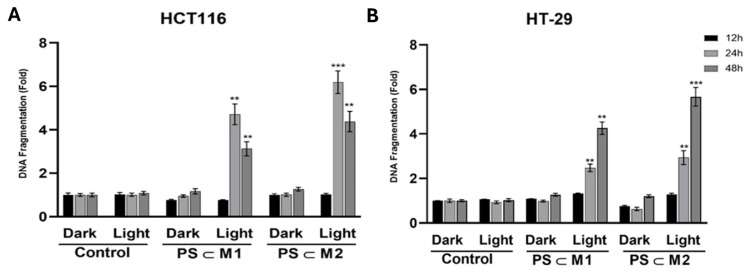
DNA fragmentation in HCT116 (**A**) and HT-29 (**B**) cells analyzed from cytosol extracts using ELISA assay. After seeding the cells for 24 h, followed by treatment, the cells were illuminated (630 nm, 75 J/cm^2^) or not. After 12 h, 24 h, and 48 h post-illumination, the cells were collected, and the level of DNA fragmentation was analyzed. Histograms are represented as a mean ± SEM of at least three independent experiments. ** *p* < 0.01 and *** *p* < 0.001.

## Data Availability

The data are contained within the article.

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
