# Peer review of "Photodynamic Therapy against Colorectal Cancer Using Porphin-Loaded Arene Ruthenium Cages"

_ijms, 2024, doi:10.3390/ijms251910847_

Round 1

Reviewer 1 Report

Comments and Suggestions for Authors

The paper “Photodynamic Therapy Against Colorectal Cancer Using 2 Porphin-Loaded Arene Ruthenium Cages” describes a very interesting work, it’s a highly complete study in which the demonstration of the anticancer action of the photoactivable tools synthetized here was performed on 2 different cell lines. A large panel of biological and biochemical techniques were used to describe the mechanism involved in the cell death photoinduced. It’s a particularly in-depth study that deserves to be published after minor corrections

1) (page 4) Please replace 12h, 24h and 48h by 12 h, 24 h and 48 h or 12, 24, and 48 h!

2) Would it be possible to group together the fig 2 and 3? Indeed, it’s unusual to see 2 figures for the same result presented differently. In addition, remove “or not” in the legend after illumination

3) the legend of figure 5 contains too many technical details, please remove those which are not necessary here but more adapted in the experimental part ( ex: ibidi µ-slides…)

 4) For the calculation of the amount of joules delivered, could you verify the value of 75J/cm2?

J = W x sec

J/cm2 = W/cm2 x sec

J/cm2 = 70 mW/cm2 x 6 min

J/cm2 = 0.07 x 360 = 25.2

Reviewer 2 Report

Comments and Suggestions for Authors

Dear the Authors

This work is very similar with previous author's paper (Ref. 30). PS (porphin) and M1 and M2 cages were the same (not new!). Only the authors tried in different cells. Therefore, it is need to make a paper again with high novelty and differences. 

1. In Introduction part, it is need to make a paragraph for this work's purpose and difference.

2. There is no control activity (PDT) of free PS (without M1 or M2) in cell viability with IC50 value. 

3. Please change the concentration unit to mictomole (not nanomole).

4. Light dose used in PDT work (75 J/cm2) is high, therefore, it is need to reduce it. 

5. In 4.4. Light Source, calculated light dose it not correct (70 mW (0.07 W x 6 min (60 s x 6)???).

6. Figure 10 should be moved to Figure 1 with related sentences in Introduction part. 

Round 2

Reviewer 2 Report

Comments and Suggestions for Authors

Dear Authors

Thanks so much for your revised version of this paper. 

I think it is need to add related sentence and reference(s) for the poor solubility of free PS (porphin) to make a clear purpose of this paper. 

Also please add concept (Figure 10 on the orginal manuscript) as a Figure 1 to understand the purpose of this paper with novelty (NOT as a Cover art only). 

And please add sentence(s) for novelty and significance of this paper work. 

Thanks.